# Surgical Management of Spinal Tuberculosis—The Past, Present, and Future

**DOI:** 10.3390/diagnostics12061307

**Published:** 2022-05-24

**Authors:** Sameer Ruparel, Masato Tanaka, Rahul Mehta, Taro Yamauchi, Yoshiaki Oda, Sumeet Sonawane, Ram Chaddha

**Affiliations:** 1Department of Orthopaedic Surgery, Jaslok Hospital and Research Centre, Mumbai 400026, India; spineram@gmail.com; 2Department of Orthopaedic Surgery, Okayama Rosai Hospital, Okayama 702-8055, Japan; tanaka0896@gmail.com (M.T.); ygitaro0307@yahoo.co.jp (T.Y.); odaaaaaaamn@yahoo.co.jp (Y.O.); 3Department of Orthopaedic Surgery, R.D. Gardi Mecial College, Ujjain 456001, India; drrahulpmehta@gmail.com; 4Department of Orthopaedic Surgery, BKL Walawalkar Medical College, Sawarde 415606, India; drsumeet166@gmail.com

**Keywords:** spinal tuberculosis, surgical management, management principles

## Abstract

Tuberculosis is endemic in many parts of the world. With increasing immigration, we can state that it is prevalent throughout the globe. Tuberculosis of the spine is the most common form of bone and joint tuberculosis; the principles of treatment are different; biology, mechanics, and neurology are affected. Management strategies have changed significantly over the years, from watchful observations to aggressive debridement, to selective surgical indications based on well-formed principles. This has been possible due to the development of various diagnostic tests for early detection of the disease, effective anti-tubercular therapy, and associated research, which have revolutionized treatment. This picture is rapidly changing with the advent of minimally invasive spine surgery and its application in treating spinal infections. This review article focuses on the past, present, and future principles of surgical management of tuberculosis of the spine.

## 1. Introduction

For centuries, Tubercle bacilli have been known to live in symbiosis with mankind. The earliest description of tuberculosis affecting humans dates back to pre-1000 BC. Hippocrates (100–300 BC) is credited for the first description of spinal tuberculosis. Although the disease has been known for a long time, the primary pathogen, Mycobacterium tuberculosis, was only recently isolated [1]. Lungs are the most common site of involvement; extrapulmonary manifestations are also fairly common. A study of bone and joint tuberculosis conducted in the United States over a 4-year period stated that the spine was the most affected region in tuberculosis of the bone (40%), followed by weight-bearing joints (hips and knees) [2].

Principles of management for spinal tuberculosis are different as it affects neurology, biology, and mechanics. The development of the Bacille Calmette–Guérin (BCG) vaccine and effective anti-tuberculous drugs have changed these principles from watchful observations to active evidence-based combination therapies of medical and surgical management. However, changing principles have led to controversies in the literature. There has been a debate regarding the indications, surgical principles, approaches, and instrumentations, to name a few, of surgical management of spinal tuberculosis. Similarly, with the advent of minimally-invasive spine surgery and percutaneous pedicle screw fixation, these horizons are rapidly changing. This review article discusses the evolution and development of the principles of the surgical treatment of spinal tuberculosis.

## 2. The Past

Spinal tuberculosis was described by Percivall Pott in 1779 as ‘that kind of palsy of lower limbs which is frequently found to accompany a curvature of the spine’ [3]. Before the development of effective anti-tubercular drugs, the mainstay of treatment was immobilization using body casts or plaster beds. Patients were admitted in sanatoria; there was an emphasis on recumbency, fresh air, and nutritious food [3], with long periods of hospitalization. The primary aim of this orthodox non-operative treatment was to achieve the stage of ankyloses in the least-disabling position. However, the results were disappointing; only 30–44% of treated patients resumed full working capacity [4,5,6]. The rest either died (30–50%) or were paralyzed and severely crippled. However, rest, fresh air, and nutritious food are emphasized even today as an integral part of treatment.

Unsatisfactory results of non-operative treatment prompted physicians to develop approaches for surgical excision of diseased bones and joints [7,8]. The general outlook regarding surgery was aptly summarized by Calot [9]: the ‘surgeon who, so far as tuberculosis is concerned, swears to remove the evil from the very root, will only find one result awaiting him—the death of his patient’. This prompted surgeons to develop ‘distant operations’ away from the site of pathology. Albee [10,11] and Hibbs [12,13] developed posterior spinal fusion to shorten the period of immobilization. However, since the primary site of pathology was not treated, the results were not optimal [7,8].

The development of anti-tuberculous drugs in the mid-nineteenth century bought about a revolution in the surgical treatment of spinal tuberculosis. It allowed surgeons to directly access and debride the site of pathology. Since there was a belief that anti-tubercular drugs did not penetrate osseous lesions, the principle was universal excisional surgery [14,15,16]. However, many reports later showed effective penetration of drugs in bony lesions including cavities and abscesses [17,18,19]. Thus, the principles of radical anterior debridement as proposed in the ‘Hong Kong’ procedure by Hodgson and Stock [14] changed to the removal of only sequestrated vertebrae or discs or the offending tissues compressing the dural tube [20,21,22]. Medical Research Council (MRC) trial results led to the development of the middle path regimen, which was put forward by Dr. Tuli to manage patients with spinal tuberculosis and neurological deficits, making the indications for surgery more selective today [21,23,24]. The MRC trials divided patients into three prospective groups. They concluded that the drug treatment group, debridement group, and debridement with anterior spinal fusion group had similar outcomes at 15 years in terms of healing [25]. However, they did not consider deformity. Long-term evaluation of these patients showed that the deformity progressed in the first two groups. Similarly, a landmark study by Oga et al. [26] concluded that Tubercle bacilli did not adhere to metals and form biofilms, making it possible to use instrumentation at the site of pathology for immediate stability and to further improve the quality of life of patients.

## 3. The Present

The development of effective anti-tuberculous treatment (ATT) and the results of numerous studies to date have revolutionized the management of patients with spinal tuberculosis over the years and formed well-defined principles today. Tuberculosis of the spine is now regarded as a ‘medical disease’ with indications of surgery being pathology causing neurological affection or deformities, compression of surrounding vital structures due to an abscess, and clinical deterioration (despite ATT). The principles of surgical management include addressing the lesion with *optimum debridement and decompression*, restoring the *stability of the spine* with adequate *bone-on-bone contact* for fusion (providing for early mobilization), and using a *surgical approach* that is most effective and least morbid to the patient coupled with *effective ATT. Optimum decompression* includes removing the offending structures causing neural compression with loose sequestered pieces of the bone and disc, and drainage of pus pockets. Radical debridement—until there are bleeding bone edges—is not recommended as it causes large anterior defects requiring reconstruction. *Bone-on-bone contact* for fusion is accomplished with strut grafts (autograft/allograft) or titanium mesh cages spanning the site of decompression. This can be done with either an anterior or posterior approach. *Immediate stability* for early mobilization is provided with instrumentation. It also promotes healing and enhances fusion rates. The pediatric spine instability score proposed by Rajasekaran, based on four ‘spine at risk signs’ (facet subluxation, retropulsion, lateral translation, and toppling sign), holds true to assess the severity of the progression of the deformity. Each radiological sign is allocated one point, with a maximum possible score of four. A score of two or more is considered unstable and requires stabilization [27]. Recently, a new objective scoring system to assess spinal instability was developed based on five main factors [28] (Table 1).

Isolated arthrodesis without instrumentation was shown to be associated with graft-related complications and the progression of kyphosis. Upadhyay reported dislodgment of the graft after anterior fusion in 10 out of 104 patients and an increase in kyphosis by 20 degrees in one year [29]. Finally, *effective ATT* is the most important pillar for successful outcomes of spinal tuberculosis. Currently, we follow the WHO recommendations of 9 months of treatment, where four drugs—isoniazid, rifampicin, pyrazinamide, ethambutol, or streptomycin—are administered in the “initiation” phase for 2 months, followed by isoniazid and rifampicin for 7 months in the “continuation” phase [30]. With the emergence of multi-drug-resistant (MDR) tuberculosis, the earlier practice of empirical treatment is now changing to evidence-based biopsy-proven ATT based on drug sensitivity patterns. In fact, a recent study [31] found 43.6% of Xpert-MTB-proven spinal tuberculosis (TB) cases to be multi-drug-resistant (MDR-resistant) to both rifampicin and isoniazid.

One of the principles of optimum surgical management is using an *approach* that is most effective and least morbid to the patient, as mentioned above. This can be achieved with either an anterior or posterior approach or a combined approach (Table 2).

In comparison to the cervical, lumbar, and lumbosacral regions, the thoracic-thoracolumbar spine is more commonly affected and the controversy regarding this approach exists for this region [32]. Since the disease usually affects anterior vertebral structures, the anterior approach is considered the gold standard [33]; this approach remains the main stay of treatment in cervical subaxial tuberculosis. It provides direct access to the pathology, allowing thorough removal of necrotic tissues and abscesses with the ease of inserting large grafts. A subtotal or complete corpectomy was conducted of the involved vertebrae to achieve neural decompression. For the purpose of reconstruction, a titanium or polyether ether ketone (PEEK) cage was installed after infusing it with an autogenous rib or iliac crest grafts. A rod screw construct completes the surgery in the form of instrumentation anteriorly (Figure 1).

Dai et al. conducted a prospective study involving 39 cases of spinal tuberculosis; they found that the therapeutic effect of the anterior approach was excellent in single-stage anterior debridement, bone grafting, and instrumentation with a very low rate of non-healing [34]. The average kyphotic deformity in their series was 13.5 degrees. Li et al. [35] also stated that the durations of surgery and blood loss in procedures conducted by the anterior approach were significantly lower than in surgeries approached posteriorly. They excluded patients who had previous thoracic surgery histories, active lung tuberculosis, or apparent presentation of spinal cord damage. They concluded that anterior transthoracic debridement and fusion were less traumatic than posterior transpedicular debridement and fusion. Their explanation for this inference was three-fold. They stated that they studied cases specifically with lesions situated in the mid-thoracic region and in the affected functional spinal units that contained two (or less than two) numbers of segments; it was not required to cut-off ribs in a quest to obtain proper exposure and good access to the lesion. A second tool that they used was the LigaSure or LigaClip vessel-closure system instead of the earlier system of isolating and ligating vertebral transverse arteries. This saved time and significantly reduced blood loss. They tackled the third hurdle of surgeon preference by involving thoracic surgeons in all surgeries. Anterior surgery is ideal in cases of young patients with no comorbidities having mid-thoracic lesions or single-level discitis and minimal kyphosis. Surgeon experience plays a major role in a successful surgery from the anterior approach. However, cases with junctional TB affecting more than two levels, patients with pulmonary lesions, and elderly obese individuals are major contraindications in this approach. Similarly, stability provided by instrumentation is questionable as it anchors onto the infected osteoporotic bone, risking translation and inadequate correction of deformities [36,37]. The combined approach has advantages of both anterior and posterior approaches, providing excellent decompression and stability. This approach starts with the patient being placed in a prone position. A midline longitudinal incision is made at the center of the diseased vertebral body and exposure is conducted to reach the posterior elements. Pedicle screws are inserted in the involved vertebrae depending on the extent of the damage. In addition, screws are inserted in vertebrae with intact pedicles at levels above and below the involved vertebrae. Correction of Kyphosis is carried out with the help of an internal fixation system. After this, the patient is moved to the right lateral position. The thoracic incision is again taken based on the center of the involved vertebrae. The tuberculous lesion is exposed before removal of the abscesses, caseous necrotic tissues, sequestrate, necrotic discs (partial corpectomy), and granulated tissues. After the preparation of a fusion bed, a bone graft is placed. Since this approach entails two surgical procedures, it is also the most morbid. When performed as a single event, it increases the surgical time and blood loss, increasing rates of infection and morbidity [38,39]. Krodel et al. [40] found blood loss to be as high as 1700 mL, which is huge in patients with poor general conditions, such as spinal tuberculosis. However, in patients with large lesions causing three-column destructions and severe deformities, a combined approach (Figure 2) is suitable [41].

Recently, only posterior or posterolateral approaches (Figure 3) have been described [42,43,44], providing access to anterior and lateral columns through extra-pleural techniques. This entails making the patient prone. The thoracolumbar median approach is used. A posterior midline incision is taken. The cord is decompressed using the posterolateral extra-pleural approach. Curettes are used to remove the necrotic material within the vertebral body and the disc; drainage of the paraspinal abscess is conducted. The defect is reconstructed from one side using a titanium mesh cage infused with an autograft. Stabilization is conducted by installing pedicle screws and rods. Fusion is conducted at levels above and below the involved level, depending on the extent of involvement. This approach has become popular since it allows for circumferential decompression of the spinal cord, can be extended proximally and distally, and provides excellent stabilization with deformity correction through a strong three-column instrumentation [45]. Thus, this approach can be used for stabilization alone, stabilization and decompression, or stabilization, decompression, and anterior reconstruction. In a meta-analysis, Liu et al. [46] concluded that the posterior approach has the same clinical efficacy but with fewer operation times, blood loss, hospital stays, and complications when compared with combined surgeries. Since it was a meta-analysis it had few limitations. The recruited studies were not randomized controlled trials and had small sample sizes. Heterogeneities among studies may distort pooled results.

*Surgery in healed TB spines*: Utilization of the posterior approach for surgery in a healed TB spine requires special mention. This usually presents with sharp angular deformities or late-onset neurological deficits. In such cases, it is difficult to approach the spine through an anterior transthoracic approach [47]. Combined approaches employ anterior corpectomy, posterior shortening and instrumentation, and anterior and posterior grafting [48,49]. Though these multi-stage techniques were initially employed to treat them, recently, various techniques, such as transpedicular de-cancellation procedures, pedicle subtraction osteotomies, vertebral column resections, and closing opening wedge osteotomies [50,51,52,53] have shown excellent comparable outcomes. Depending on the severity of the angular deformity, either technique is undertaken. Achieving deformity correction by restoring the anterior column height might stretch the sensitive spinal cord; hence, the above techniques involve taking out wedges (with the base of the wedge posterior) along with compression maneuverers to correct deformities. Closing opening wedge osteotomies is particularly useful in tuberculosis, which may involve the collapse of up to three vertebral bodies. A pure closing technique might produce severe kinking of the cord [54]. Hence, in such surgeries, the anterior column is reconstructed with strut grafts/vertebral spacers filled with autografts. Rajasekaran et al., in a series involving 17 patients, reported that the average kyphosis improved from 69.2° preoperatively to 32.4° postoperatively by closing the opening wedge osteotomy [46]. Researchers employed a novel technique of apical spinal osteotomies through a posterior approach in rigid severe kyphoscoliosis and found excellent results, validating its use in children under 14 years of age [55].

Thus, each approach has its advantages and disadvantages. Any approach utilized is ultimately acceptable, provided that the principles of surgical treatment of spinal tuberculosis are adhered to. Mehta and Bhojraj, in 2001, created a new system for classifying spinal TB based on magnetic resonance imaging (MRI) findings (Table 3). This study can be used as a basis for decisions regarding approaches to be used for patients with spinal tuberculosis. Anterior debridement and strut grafting is conducted for patients with stable anterior lesions and no kyphotic deformity. Global lesions with kyphosis and instability are better treated with posterior instrumentation surgery in addition to an anterior strut graft. Patients who suffer from global lesions and have high surgical risks for transthoracic surgery due to medical comorbidities and probable anesthesia-related complications are considered suitable candidates for decompression posteriorly. The transpedicular route is used to approach the anterior aspect of the cord in addition to posterior instrumentation. Finally, only a posterior decompression is conducted for patients with isolated posterior lesions [56]. This study mainly used the Hartshill rectangle for posterior stabilization; however, the principles remain the same whether we use pedicle screws or the Hartshill rectangle for fixation.

## 4. The Future

Spine surgeons around the world endeavor to provide the best clinicoradiological outcomes with the least morbidity. This thought process has ushered in the era of minimally invasive spine surgery (MISS). MISS is any surgery that achieves the aim of a surgical procedure with minimal collateral damage to the surrounding normal tissue, using natural musculoskeletal planes and the smallest possible footprint. The advantages of MISS (Table 4) in degenerative spine pathologies are well established today.

It is proven to cause less blood loss, less post-operative pain, smaller incisions, fewer post-operative stays, and early returns to work; thus, it has better patient outcomes [57,58]. The spectrum of MISS for application in tumors, deformities, and trauma is slowly widening [59,60], and the same goes for infections (particularly tuberculosis). A recent meta-analysis by Wang et al. [61] comparing anterior versus posterior instrumentation for spinal tuberculosis found blood loss in the range of 385–1100 mL. They also found that the average duration of hospital stay ranged from one week to a month and a half for either approach. It would be worthwhile if these statistics could be brought down for better functional outcomes for patients using MISS techniques. The earlier applications of MISS were restricted to percutaneous biopsies and abscesses drainage. Ashizwa et al. [62] performed percutaneous transpedicular biopsies and found 92% accuracy without significant complications. Similarly, if abscesses fail to resolve with chemotherapy or cause pressure symptoms, percutaneous drainage with transforaminal or posterior full endoscopic techniques help in immediate symptom resolution as well as in obtaining representative samples for laboratory studies for any changes in drug regimens [63]. Video-assisted thoracoscopic surgery (VATS) has been used for biopsies and decompressions supplemented with posterior percutaneous pedicle screw fixation [63,64]. With the success of minimally invasive anterior and anterolateral approaches to the lumbar spine in treating degenerative pathologies, these are increasingly being used to treat lumbar and lumbosacral spinal tuberculosis.

Xu et al. [65], in a comparative study of 83 patients affected with lumbar tuberculosis, found direct lateral interbody fusion (DLIF) with posterior percutaneous screw fixation to have less intraoperative blood loss and less hospital stay as compared to traditional anterior extra-peritoneal debridement and posterior internal fixation, contributing to early rehabilitation of patients. Similarly, Du et al. [66] conducted a retrospective study and found oblique lumbar interbody fusion (OLIF) with percutaneous pedicle screw fixation to have the advantages of less surgical trauma, faster recovery, and shorter fusion times with similar clinical outcomes when compared with traditional posterior transforaminal or transpedicular approaches, and pedicle screw fixation for single-segment lumbar tuberculosis. Both DLIF and OLIF techniques were shown to offer thorough anterior debridement and fusion for up to two levels of involvement, supplemented with percutaneous fixation posteriorly. The evolution of expandable cages with variable footplates has helped achieve excellent kyphosis correction and lost height due to the disease process by placement through narrow corridors. Recently, these techniques have also been extrapolated for the treatment of thoracic tuberculosis. A case report by Deng et al. [67] found extreme lateral interbody fusion (XLIF) with percutaneous pedicle screws feasible for the treatment of thoracic tuberculosis, allowing debridement, discectomy, and interbody fusion under direct vison, improving patient prognosis. However, there have been concerns regarding increased radiation exposure while inserting percutaneous screws [68]. This has been taken care of with the help of computer-assisted navigation systems. Jiang et al. [69], in a retrospective comparative study of 33 patients with lumbar tuberculosis treated with DLIF and navigation-assisted fluoroscopy, found a significant reduction in radiation exposure with comparable operative time, blood loss, length of stay, and complication rates with non-navigation-assisted fluoroscopy and DLIF.

The use of endoscopes for treating spinal pathologies is presently in vogue and there are reports of treating spinal tuberculosis with these tools as well. Upper cervical—as well as sacroiliac joint tuberculosis—are difficult lesions to approach. Liu et al. [70] treated 17 patients with upper cervical tuberculosis with endoscopic-assisted anterior cervical debridement with posterior fixation and fusion and found it to be an effective and feasible surgical technique. Similarly, Guan et al. [71] conducted percutaneous endoscopic debridement and lavage in seven patients with sacroiliac tuberculosis. They concluded that this technique causes less tissue trauma and quick post-operative recovery. Complex screw placement, osteotomy, and correction maneuvers in post-tubercular deformities markedly increase the risk of spinal cord ischemia injury, leading to a postoperative neurological deficit ranging from 4 to 21.2% [72,73]. With evolving trends of MISS techniques, one cannot neglect the safety of these patients, where intraoperative neuromonitoring (IONM) has served as a boon [74,75]. IONM provides real-time feedback regarding the functioning across the operated segment via multimodal monitoring, allowing the surgical team to obtain a reasonable deformity correction without endangering the spinal cord. In fact, planning such complex osteotomies can now be done preoperatively. Hu et al. [76], in a series of 18 patients, demonstrated the role of image processing and planning software such as SURGIMAP (Nemaris, Inc.) in determining preoperatively the level and type of osteotomy to achieve the best surgical correction. We believe that this is just the beginning of applying MISS techniques to treat spinal tuberculosis; soon, most future cases will become amenable to them, further improving clinicoradiological outcomes.

## 5. Conclusions

Tuberculosis of the spine has affected humans for centuries. With advancements in scientific technology, treatment modalities and management strategies are ever-changing. Due to the rise in immigration, tuberculosis is now endemic in most parts of the world; it is necessary for astute spine surgeons to be aware of present surgical principles and future directions. Initially, watchful observations were the treatments of choice due to dismal surgical outcomes. ‘Distant Operations’ were not successful either. The development of the Bacille Calmette–Guérin (BCG) vaccine and effective anti-tuberculous treatment(s) (ATT) were landmark events that revolutionized the management of spinal tuberculosis centered on evidence-based medicine. These, along with the results of Medical Research Council (MRC) trials, formed the principles of effective management for spinal tuberculosis today. The overview gradually changed from aggressive surgical debridement to selective surgical indications affecting neurology and stability of the spine. Tuberculosis is now a ‘Medical’ disease and effective ATT has been the most important pillar for successful outcomes of spinal tuberculosis. The WHO recommends 9 months of treatment, with the following drugs—isoniazid, rifampicin, pyrazinamide, ethambutol, or streptomycin—being administered in the “initiation” phase for 2 months, followed by isoniazid and rifampicin for 7 months in the “continuation” phase. Numerous studies have also documented the feasibility of using implants in patients with Koch’s spine, providing immediate stability and improving functional outcomes for patients. Similarly, surgical approaches have been subjects of controversy and have changed throughout the years. When radical operations were in vogue, anterior surgeries were preferred. The anterior approach remains the mainstay of treatment for subaxial cervical spine pathologies whereas manubrium splitting, transthoracic, transdiaphragmatic, and retroperitoneal approaches enable us to access cervicothoracic, thoracic, thoracolumbar, and lumbar pathologies, respectively. However, the familiarity and better clinical outcomes of the posterior approach have prompted more spine surgeons to treat these pathologies posteriorly. These can be transfacetal, transpedicular, costotransversectomy, and extended approaches for anterior column reconstruction (LECA). Minimally invasive spine surgeries have proven to be successful for degenerative pathologies due to smaller incisions, fewer infections, and faster recovery times. These techniques are increasingly being used in spinal tuberculosis with selective indications and good success rates. Navigation and intraoperative neuromonitoring have significantly decreased complication rates associated with the correction of complex, severe, and rigid post-tubercular deformities. Globally, spine surgeons are striving for better patient outcomes; they have made it possible to achieve clinically successful outcomes today. It is said that ‘History repeats itself’; however, we hope that in the case of tuberculosis it does not. The emergence of multi-drug-resistant (MDR) and extreme drug-resistant (XDR) tuberculosis is a cause of worry and speculation. We hope that it does not change the effective treatment principles formed today; if it does, we will find an effective solution to treat it as we did in the past.

## Figures and Tables

**Figure 1 diagnostics-12-01307-f001:**
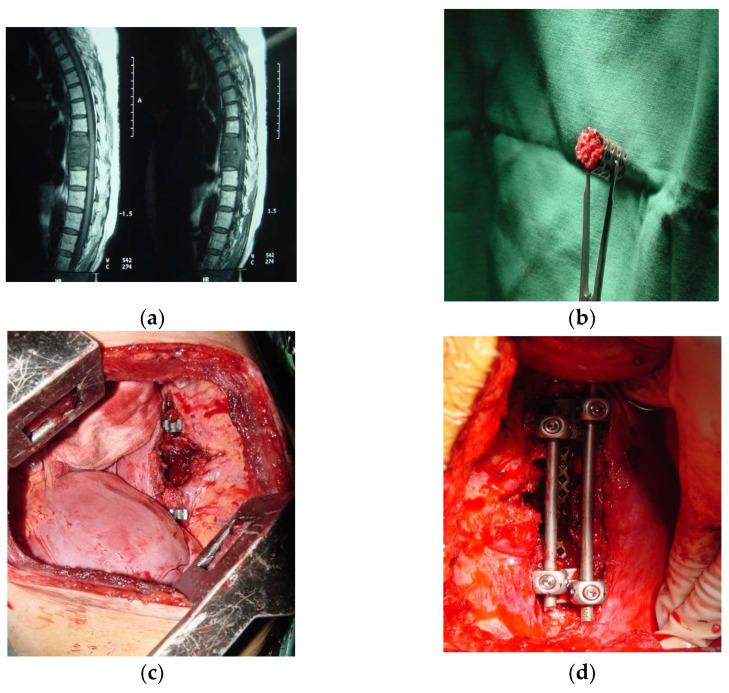
(**a**) Shows a MID-thoracic tuberculous lesion on MRI; (**b**) cylindrical mesh cage with infused bone graft; (**c**) shows an intraoperative picture with a prepared site for cage placement; (**d**) shows the anterior fixation with rods and a mesh cage; (**e**,**f**) shows post-operative radiological images of anterior transthoracic surgery and fusion with a mesh cage.

**Figure 2 diagnostics-12-01307-f002:**
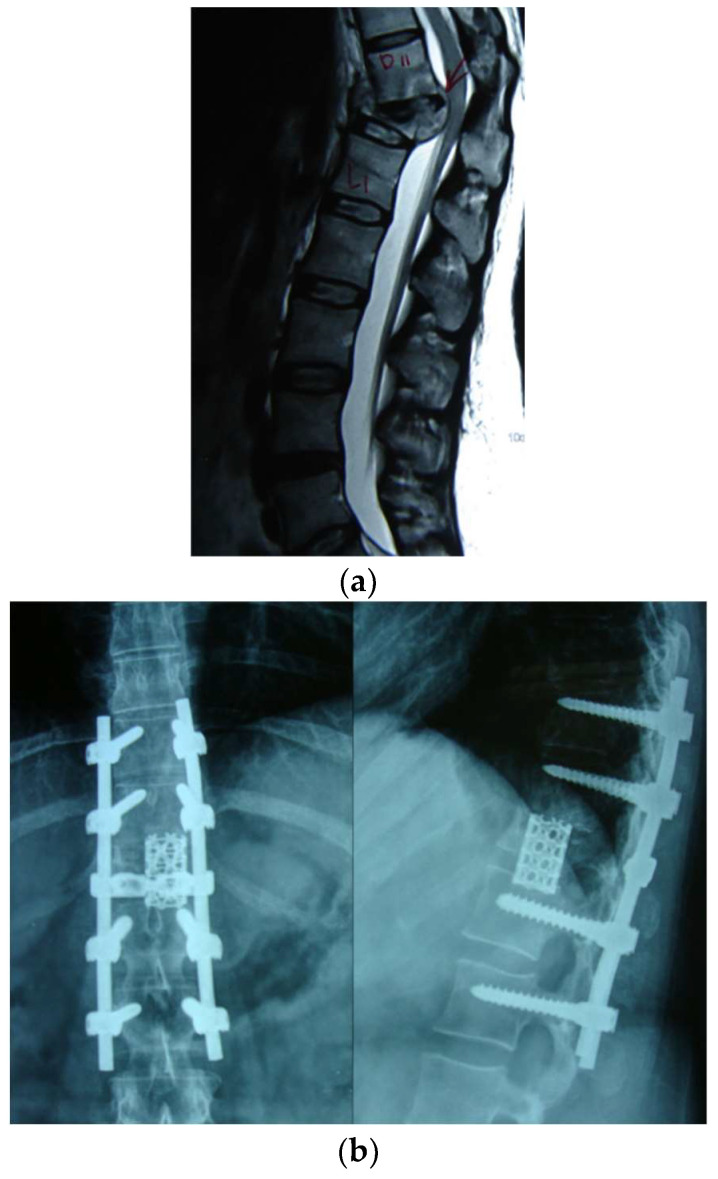
(**a**) Shows collapsed D12 vertebra with an abscess compressing the spinal cord; (**b**) shows front and back surgeries with anterior reconstruction, with a cylindrical mesh cage in addition to the posterior decompression and stabilization with pedicle screws.

**Figure 3 diagnostics-12-01307-f003:**
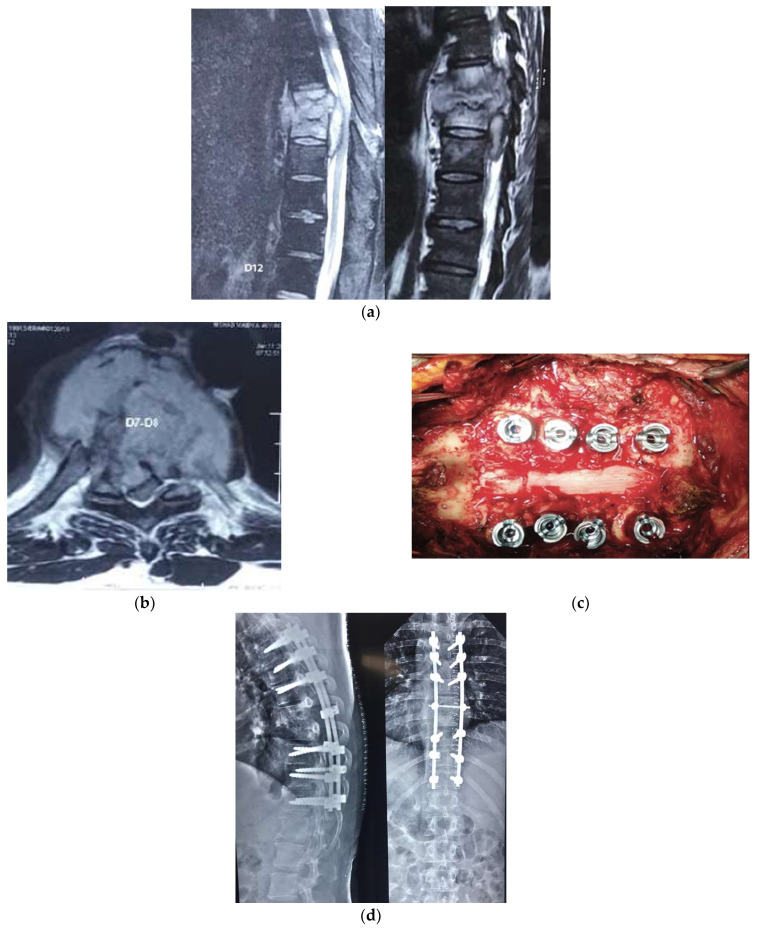
(**a**) Shows tubercular abscess on Sagittal cuts of the MRI; (**b**) shows the axial cut of the lesion; (**c**) shows the intraoperative image of the posterior decompression and stabilization; (**d**) shows the post-operative X-ray of the posterior construct comprising pedicle screws, rods, and the cross-connector.

**Table 1 diagnostics-12-01307-t001:** A validated score to evaluate spinal instability to assess surgical candidacy in active spinal tuberculosis.

Age < 15 years	1
Cervicothoracic/thoracolumbar	1
Deformity > 30 or DAR > 15	2
vertebral body loss–segmental ratio > 0.5	2
Spine at risk sign	3
Total	9

A score of zero or 1 is considered stable; 2 is potentially unstable and requires careful monitoring; and 3 and above is definitely unstable and requires surgical stabilization.

**Table 2 diagnostics-12-01307-t002:** Surgical approaches.

Anterior:	
Anterior retropharyngeal	Subaxial cervical spine
Manubrium splitting	Cervicothoracic junction
Transthoracic	Mid-thoracic spine
Transdiaphragmatic	Thoracolumbar junction
Retroperitoneal	Lumbar spine (L1–L4)
Anterior laparotomy	Lumbosacral junction
**Posterior:**	
Transfacetal	
Transpedicular	
Costotransversectomy for anterolateral decompression	
Extended posterior versatile approach for Anterior column reconstruction (LECA)	

**Table 3 diagnostics-12-01307-t003:** Magnetic resonance imaging classification of spinal tuberculosis—Mehta and Bhojraj [53].

Group A	Group B	Group C	Group D
Patients with stable anterior lesions and non-kyphotic deformity, which are managed with anterior debridement and strut grafting.	Patients with global lesions, kyphosis, and instability, and are managed with posterior instrumentation using a closed-loop rectangle with sublaminar wires plus anterior strut grafting.	Patients with anterior or global lesions along with high operative risks for transthoracic surgery due to medical comorbidities and probable anesthetic complications. Therefore, these patients undergo posterior decompression with the anterior aspect of the cord being approached through a transpedicular route and posterior instrumentation performed using a closed-loop rectangle held by a sublaminar wire.	Patients with isolated posterior lesions that only need a posterior decompression.

**Table 4 diagnostics-12-01307-t004:** Advantages of minimally invasive spine surgery (MISS) over conventional surgery.

	MISS	Conventional Technique
Blood Loss	Less blood loss	More blood loss
Post-operative pain	Less immediate post-operative pain	Comparatively more post-operative pain
Hospital stay	Shorter hospital stay	Longer hospital stay
Return to work	Early return to work	
Cost	Reduced indirect cost	
Radiation to surgeon	More	Less

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
