# Peer review of "Surgical Management of Spinal Tuberculosis—The Past, Present, and Future"

_diagnostics, 2022, doi:10.3390/diagnostics12061307_

Round 1

Reviewer 1 Report

  1. Some grammatical error, please check.
  2. “This manuscript…” should be changed to “This review…”
  3. The authors should discuss about the timing of anti-tuberculosis drugs regimens.
  4. The authors should grouping the techniques of surgeries.
  5. If possible, the bias of each included study should be evaluated.

Author Response

Response to Reviewer 1 Comments

Point 1: Some grammatical error, please check

Response 1: Grammatical errors cross checked as advised

Point 2: “This manuscript…” should be changed to “This review…”

Response 2: ‘This Manuscript’ has been changed to ‘this review’ at 2 places as advised- one in abstract and other in introduction.

Point 3: The authors should discuss about the timing of anti-tuberculosis drugs regimens

Response 3: The timing of anti –tuberculous drug regimens have been added in ‘The Present’ paragraph while discussing ‘Effective ATT’ as one of the principles of treatment as recommended.

Point 4: The authors should grouping the techniques of surgeries.

Response 4: We have added a table (Table 2) based on grouping of techniques of surgeries. This table enlists the different techniques used via anterior and posterior approaches based on the level of involvement of spinal column as well.

Point 5: If possible, the bias of each included study should be evaluated

Response 5: Whereever possible, bias of studies in study is evaluated and included in the text.

Reviewer 2 Report

The paper is a narrative review of the surgical treatment of spinal TB . The paper is comprehensive & covers all aspect of the surgical care of TB .  Congratulations to the authors on a well written paper .

Comments ;

1) Abstract – a)line 2 - However, with increasing immigration, it would be convenient to say that it is prevalent throughout the globe  . The sentence needs to be rephrased & the word convenient is not appropriate .

  1. b) There is no literature evidence that BCG has revolutionized care  in Spinal TB .  Things that has mattered is  diagnostics , anti TB drugs &  advances in surgical management .

2) introduction : adequate

3) The past –

Line 51- The average 51 time of hospitalization varied between 1 to 5 years.   – can be deleted or rephrased ( long periods of hospitalisation )

MRC trails proved that the outcome was same in all groups in terms of healing .   However they did not take deformity into considerations . Long term evaluation of Pts who were included in MRC trials have shown that deformity progresses in the first 2 groups

Middlepath regimen was put forward by Dr Tuli for the management of spinal Tb with neurological deficit .

4) The Present _

 Need better images – Fig 1

Anterior approach still remains the mainstay of treatment in cervical subaxial TB

Posterior approach could be either stabilisation alone, stabilisation & decompression . Stabilisation ,decompression & anterior reconstruction

Pgaer 6 – line 182 – delete the sentence -However, the disadvantage is 182 that decompression and reconstruction of anterior column is indirect

 There has been 2 publications on instability scores & they need to included .

Fig 3 – legends do not clearly indicate what was performed .

Table 3 – is mainly used along with hartshill rectangle .  It would be better to mention that the priciples remain the same whether we use hartshill or pedicle screws .

Future & conclusions adequate -

Reviewer 3 Report

In this review article, Sameer Ruparel et al. summarized the past, present 
and future principles of surgical management of tuberculosis of the spine. Overall the manuscript is well organized and written and would be interesting to both surgeons and infectious disease experts. Following concerns need to be addressed in the revision. 

  • Table 1 and 2. It's unnecessary to present Indications for surgical management or Effective surgical principles as tables. Text will works.
  • Table 4. It would be more informative to compare Minimally Invasive Spine Surgery (MISS) with traditional surgery in the table. 

Round 2

Reviewer 1 Report

In conclusion section and based on the evidences, it is better if the authors summarize/recommend the antitubeculous drugs (type, timing) and the surgical techniques.

Author Response

Point 1: In conclusion section and based on the evidences, it is better if the authors summarize/recommend the antitubeculous drugs (type, timing) and the surgical techniques.

Response 1: The regimen of anti tuberculous drugs and surgical techniques have been summarized in conclusion section as advised.